# Cardiotoxicity of Electronic Cigarettes and Heat-Not-Burn Tobacco Products—A Problem for the Modern Pediatric Cardiologist

**DOI:** 10.3390/healthcare11040491

**Published:** 2023-02-08

**Authors:** Alina-Costina Luca, Alexandrina-Ștefania Curpăn, Alin-Constantin Iordache, Dana Elena Mîndru, Elena Țarcă, Florin-Alexandru Luca, Ioana-Alexandra Pădureț

**Affiliations:** 1Sfânta Maria’ Emergency Children’s Hospital, 700309 Iași, Romania; 2Department of Pediatric Cardiology, Faculty of Medicine, Gr. T. Popa’ University of Medicine and Pharmacy, 700115 Iași, Romania; 3Department of Biology, Faculty of Biology, “Alexandru Ioan Cuza” University of Iași, Bd. Carol I, 20A, 700505 Iași, Romania; 4Department of Mother and Child Medicine–Pediatric Cardiology, “Grigore T. Popa”, University of Medicine and Pharmacy of Iasi, 16 Universitatii Str., 700115 Iași, Romania; 5Department of Surgery II—Pediatric Surgery, Grigore T. Popa’ University of Medicine and Pharmacy, 700115 Iași, Romania; 6Department BMTM, “Gheorghe Asachi” Technical University of Iasi, 700050 Iaşi, Romania

**Keywords:** cardiotoxicity, e-cigarettes, heat-not-burn, adolescents, pediatric cardiology

## Abstract

Electronic nicotine delivery systems (ENDS) have become increasingly popular among adolescents, either as an alternative to conventional cigarettes (CCs) or as a newly acquired recreational habit. Although considered by most users as a safer option for nicotine intake, these devices pose significant health risks, resulting in multisystem damage. Heat-not-burn products, which, unlike ENDS, contain tobacco, are also alternatives to CCs that consumers use based on the idea that their safety profile is superior to that of cigarettes. Recent studies in the USA and EU show that adolescents are particularly prone to using these devices. Pediatric cardiologists, as well as other healthcare professionals, should be aware of the complications that may arise from acute and chronic consumption of these substances, considering the cardiovascular damage they elicit. This article summarized the known data about the impact of ENDS on the cardiovascular system, with emphasis on the pathophysiological and molecular changes that herald the onset of systemic lesions alongside the clinical cardiovascular manifestations in this scenario.

## 1. Introduction

Traditional cigarettes have been replaced, especially in young consumers, by a variety of electronic devices that mimic the smoking experience. However, the overall prevalence of cigarette smoking has remained considerably high amongst both women and men, with the higher prevalence being in Asia followed by Europe and USA; however, when it comes to smoking rates in adolescents, the highest rate is in Europe and South America, which have seen an increasing trend, especially when it comes to electronic cigarettes [1].

Electronic nicotine delivery systems (ENDS, which include vapes, e-hookahs, e-pipes, and e-cigars) rely on aerosolizing nicotine containing flavored liquids. ENDS have been promoted as an alternative to traditional smoking and portrayed as helping with smoking cessation [2]. The main components of an electronic cigarette include a mouthpiece, a cartridge, a heating element/atomizer and a battery (a general structure of ENDS can be observed in Figure 1). The liquid is stored in the cartridge and then supplied to the atomizer containing a heater that heats up when the battery is turned on. Electronic devices for nicotine consumption run on batteries, and the elimination of aerosols is based on a process of gradual heating and not combustion. The heating mechanism behind electronic cigarettes poses the risk of exposure to toxic contaminants such as cadmium, chromium, lead, nickel, and silicates [3,4]. Even if concentrations are generally low, nanoparticles of heavy metals are of concern because they are easily absorbed. Electronic cigarettes are also capable of producing highly reactive free radicals [3,5]. The circulating medium for nicotine in electronic cigarettes is propylene glycol or glycerol, which by thermal degradation leads to the formation of carbonyl compounds such as acrolein [6] with vasoconstrictor potential, but which do not induce significant changes in cardiovascular parameters or increase oxidative stress in the absence of nicotine, indicating the latter as the main mediator of oxidative dysfunction [7,8].

The new alternative to traditional smoking that is currently taking over the market includes heat-not-burn products (HTPs), which are advertised as being less harmful than conventional cigarettes because they heat the tobacco at a much lower temperature than the burning one, therefore releasing less harmful substances. The heat-not-burn devices are similar to the e-cigarettes in the sense that they are comprised of a tobacco stick or heatstick, a heating holder that is powered by battery and a charger. The heating aspect alongside the fact the smoke is claimed to be less harmful to the passive smoker allows these devices to pass smoking regulations worldwide and makes them more attractive to the smoking population [9]. Although promoted as an alternative to conventional smoking, HTPs may promote dual or poly-tobacco use among the young population [10].

The harmful effects of conventional cigarettes have long been studied, and more than 7000 chemicals, such as arsenic, benzene, heavy metals, and carbon monoxide have been identified with lasting immunomodulatory effects on the lungs [11]. Even if ENDS are relatively newer to the market, a sum of harmful effects have also been related to their use both in vitro and in vitro, such as inflammation, allergic airway inflammation in asthma models, tissue damage [12], and, in humans, acute eosinophilic pneumonia and lipoid pneumonia [13,14]. It has also been shown recently that acute ENDS exposure in healthy individuals increases aortic stiffness, blood pressure, and airway resistance [15,16].

Electronic nicotine delivery systems, or e-cigarettes as they are colloquially known, have been described as posing a significant potential risk on the developing brain as well as on people living with severe mental illnesses [17]. A study in that direction has illustrated that this specific demographic has a higher mortality rate at a young age compared to other demographic groups in the same cohort study mainly due to the complications associated with tobacco use, such as cardiovascular disorders, cancers, and various lung issues [18,19].

The nicotine concentration should reach no more than 20 mg/mL (the maximum concentration accepted by the EU according to directive 2014/40/EU of the European Parliament and the European Union Council). However, studies showed that there are serious discrepancies between the nicotine concentration indicated on the label, and that found by laboratory tests [20]. Advertised nicotine concentration in the liquid pod is not necessarily the same as in the vapor; furthermore, the gaseous phase may contain dangerous substances not reported on the packaging, such as ultrafine particles of 1,2-propanediol, 1,2,3-propanetriol, and diacetin [21]. Electronic nicotine delivery devices contain various flavors in the solvent, which influence the pathogenic potential of inhaled vapors [22,23]. One category of devices that operate with subohmic atomizers, generating large amounts of volatile aldehydes with each release of aerosols, is associated with increased carcinogenic risk [24] and requires strict regulation.

Among American high school students, between 2011 and 2018, the percentage of those who used electronic cigarettes increased from 1.5% to 20.8%, with a marked increase from 11.7% in 2017 to 20.8% in 2018 [25]. In Georgia, Italy, and Romania the prevalence in the age group 11–17 years reached 12.4% in 2017, 18.3% in 2018, and 10.7% in 2018, respectively, doubling their values within 3–4 years of monitoring [26]. Another interesting aspect uncovered by the study of Tarasenko et al., is a positive correlation between pocket money and the use of ENDS—the larger the amount of money adolescents had at their disposal, the higher the probability for them to use ENDS. Chadi et al. conducted a literature review and uncovered the fact that, according to a series of observational studies performed in the US and Europe, marijuana consumption is concurrent or subsequent to ENDS use by adolescents [27,28]. Adolescents with mental health issues are at a higher risk of developing nicotine addiction [29]. According to Riehm et al., young people with internalizing and externalizing types of problems presented a higher probability of cigarette consumption. Externalizing issues proved to be an independent predictor of tobacco use in all the available delivery forms, whereas the internalizing type was correlated with ENDS alone [29].

Multiple worldwide survey studies have analyzed the prevalence of ENDS use among youth populations, both smoking and nonsmoking, and have illustrated that current smokers were more likely to use ENDS and the current use of electronic smoking devices is on the rise globally [30]. The long-term effects of e-cigarettes are still under investigation, pushing international health organizations to regulate their selling and use, as they are often marketed in a way to appeal to younger population either by the use of celebrities or attractive flavoring [31]. Another factor that influences ENDS consumption are social circles, similar to the conventional cigarettes. A study conducted on 452 undergraduate students from California State University has illustrated that among the study’s group, 82.5% of them reported using ENDS when with friends [32]. The highest number of users have been reported to be of Asian/Pacific Islander and Hispanic/Latino ethnicity. ENDS use has also been suggested as an alternative to smoking as part of nicotine-replacement therapy for surgical patients for whom cigarette smoking can increase the risk for cardiac, respiratory, and wound-related complications [33]. When it comes to young cancer survivors, a survey conducted in the USA have reported that almost 12% of young cancer survivors were currently using e-cigarettes [34], whereas almost up to 22% have reported dual use of conventional cigarettes and ENDS [34,35]. Tobacco use can impact prognosis and response to cancer therapy [36].

Unlike ENDS, the HTPs actually contain tobacco, but, instead of setting it alight, the device uses high-temperature heating, which is distributed uniformly around and through the cigarette by a central ceramic blade that pierces the cigarette. Both in vivo and in vitro studies have shown that this type of cigarette exhibits a deleterious effect on the cardiovascular system [37,38].

Cardiotoxicity is a structural and functional cause of heart damage caused by different classes of therapeutic agents, atmospheric, or food factors and is one of the most common causes of therapeutic failure, withdrawal, or disapproval of drugs, and increased incidence of cardiovascular disease unrelated to individual risk factors [39,40]. Cardiotoxicity can be irreversible, as in situations where necrosis or apoptosis of the myocardial cells [41], or reversible, as in the case of short-term nicotine product consumption or even some of the effects of prolonged smoking in heavy smokers [3,42]. The aim of this article is to explain the cardiovascular effects of ENDS and HTPs and summarize the molecular basis for the tissue damage caused by ENDS and HTPs, forecasting the long-term effects on the cardiac system, as well as possible acute complications.

## 2. Molecular Basis of the Cardiovascular Damage Associated with ENDS and HTPs

The pathway of nicotine begins when it is carried into the lungs where it is absorbed into the pulmonary venous circulation. From there, nicotine enters the arterial circulation and rapidly transitions from the lungs to the brain where it binds to the nicotinic cholinergic receptors, opening the neuron to the entry of calcium or sodium cations [43].

Recent research shows that aldehydes are by-products of the thermal degradation of flavorings used in the e-liquid. Cinnamaldehyde, a substance which gives a cinnamon-like flavor is associated with lung injury. The addition of cinnamon-flavored liquid to the cell culture of human pulmonary fibroblasts resulted in a significant increase in IL-8 and IL-6 compared to the control culture, in contrast to tobacco-flavored liquid, whose influence on the secretion of the same interleukins did not reach a significant level [12]. Interleukin-8, although mainly a granulocyte chemoattractant, is involved in the adhesion of monocytes to the vascular endothelium in the early stages of atherosclerosis, and its elevated levels are associated with coronary heart disease [44]. A concomitant increase in IL-8 and IL-6 levels is associated with hypercoagulant status and decreased fibrinolysis rate [45], as can be observed in Figure 2.

Several studies investigate the association between inflammation and e-cigarettes, especially the effects on the interleukin-6 and interleukin-8. One study analyzing the exposure of small airway epithelial cells to nicotine-free and 2.4% nicotine e-cigarettes noted a significant increase of IL-6 in the nicotine-free cell group after 24 h from exposure, whereas the nicotine-containing e-vapor did not cause a significant increase [46]. Another study using CD14+-isolated monocytes, both immature and mature, has noted an enhanced production of IL-6, whereas IL-8, IL-10, and IL-12p70 were unaffected [47]. However, a nationwide study conducted in the US on 7130 participants varying from never-smokers to active smokers has noted no difference between exclusive e-cigarette users and nonusers, and levels were lower in exclusive e-cigarettes users relative to exclusive smokers [48].

Usage of electronic cigarettes is associated with increased oxidative stress, inflammatory phenomena, endothelial injury, platelet activation, and sympathetic stimulation [49,50]. Arterial stiffness measured by changes in pulse rate and oxidative status reflected by malondialdehyde plasma concentrations increased moderately in e-cigarette users compared to tobacco smokers, but significant in terms of cardiovascular risk [51].

The harmful effects of ENDS on the cardiovascular system are under continuous research within the studies. but are unfortunately largely restricted to animal models in terms of cardiopulmonary toxicity. E-cigarette users have a higher chance of myocardial infarction and aortic stiffness, leading to a higher risk of cardiovascular events [52]. Recent studies illustrate that the exposure to e-vapors can delay ventricular repolarization in humans and chronically induce systolic dysfunction and atherosclerosis in animal models [53,54]. One study on mice has observed a dramatic decrease in heart rate in male mice, greater chronotropic sensitivity to aerosols, prolonged QT during exposure concomitant with gross bradycardia and increased ventricular premature beats, with the flavored counterparts had similar effects, but of smaller caliber [55]. Another study analyzing the effect of flavored e-vapors on cardiac electrophysiological health revealed several potential toxicity markers such as prolongation of repolarization, hERG blockage, sympathovagal imbalance, and action potential alternans [56]. Both ENDS and CCs lead to oxidative stress under the form of DNA damage, ROS generation, and apoptosis and cytotoxicity is often minimized by antioxidants [57].

The exacerbation of the cardiovascular risk associated with the use of electronic cigarettes is also based on molecular mechanisms. MicroRNAs (miRs) are noncoding RNA molecules, with a role in the posttranscriptional regulation of gene expression by interaction with messenger RNA [58]. MicroRNAs act as intercellular messengers, being involved in physiological and pathological processes. The use of electronic cigarettes causes a decrease in plasma levels of miR-29b and miR-455, which regulate metalloproteinase activity and an increase in the level of miR-155, which triggers and maintains inflammatory processes [59,60]. Early changes in miR expression after exposure to vapors from electronic cigarettes are the most conclusive evidence of their cytotoxic effect. Of great interest would be the evaluation of cardiac microRNA expression, specifically in e-cigarette consumers [59]. Moreover, Singh et al., evaluated the miRNA profile differences in exosomes between nonsmokers and smokers, depending on the type of products they use: e-cigarettes, water pipes, and CCs. E-cigarette users show increased levels of miR-21-5p, a molecule which is tied to cardiac fibrosis, heart failure, and hypertension [61,62], when compared to nonsmokers.

miR126 was found to be overexpressed in normal human bronchial epithelial cell cultures exposed to electronic cigarette condensate [63]. miR126, expressed mainly in the heart, lungs, and digestive tract, exerts its effects by regulating vascular endothelial growth factor (VEGF) and fibroblast growth factor (FGF) pathway activity [63] and is therefore responsible, among other processes, for vascular aging and angiogenesis and has proven a highly sensitive and specific marker for acute myocardial infarction (AMI) [64].

More studies are necessary in order to evaluate the levels of serum and exosomal miRNAs as a response to ENDS exposure and to establish if there is (and what is) the causal relationship between the use of ENDS, the change in the microRNA profile, and the risk of cardiovascular events in people who did not use CCs before switching to ENDS and who do not have any other predisposing factors for cardiovascular impairment.

The impact of nicotine consumption delivered by electronic devices also translates into vitamin E depletion [65], which is more pronounced in women receiving contraceptive treatment [60,66]. Vitamin E is a fat-soluble compound with anti-inflammatory valences, involved in modulating the cellular response to external aggression whose cardioprotective role has been demonstrated by the inverse association between plasma vitamin levels and heart damage [67].

A statistically significant association between volatile products from electronic nicotine delivery devices and the risk for AMI has been demonstrated [68] by a significant number of clinical and preclinical studies that evaluated the effects of both acute and chronic exposure to toxic compounds from ENDS [69,70]. For the pediatric cardiology field, it would be of immense importance to evaluate the risk for cardiovascular events considering the family history, and the individual susceptibility for the development of cardiac complications associated with tissue changes induced by electronic cigarettes, especially in young patients with previous cardiac comorbidities.

The most harmful component of e-cigarettes are the aerosols that contain a variety of fine and ultrafine particles which in turn have the potential to trigger different cardiovascular events, and, furthermore, can promote the progression of pulmonary and cardiovascular diseases [71]. Moreover, the use of ENDS on a daily basis has been suggested to be associated with a shift toward the sympathetic predominance in terms of cardiac autonomic balance, which in turn increases the hemodynamic stress and poses a risk for endothelial dysfunction, coronary spasms, left ventricular hypertrophy, and arrhythmias [72].

In cases of HTPs use, the oxidative stress and endothelial dysfunction are caused by numerous toxins, among which acrolein and benzene have been isolated. Platelet activation markers have also been linked to HTPs, and it is considered that the same mechanisms involved in CC exposure apply here [73,74]. Toxic compounds from HTPs include tar, particulate matter, ROS, and nitrosamines, which, despite their relative lower concentration compared to that of CCs, still pose a significant threat for the cardiovascular function [75,76].

Some studies suggest that this type of tobacco product emits significantly lower concentrations of tar, carbonyls, carbon monoxide, free radicals, and nitrosamines; however, this does not eliminate the risk of developing smoking-related diseases [77], as one study identified that although the concentration levels of TSNAs (carcinogenic compounds in cigarettes) were significantly lower in HTPs compared to conventional cigarettes, their transfer rate were slightly higher [78].

A systematic review analyzing the effects of HTP products on cardiovascular health have reported significant reductions in inflammation, oxidative stress, lipid metabolism, platelet function, and endothelial dysfunction, all biomarkers of exposure related to pathways involved in the cardiovascular health [79].

## 3. Pathophysiology of Cardiac Toxicity in ENDS and HTPs Usage

Numerous studies have compared e-cigs to traditional smoking and the more newly introduced HTP cigarettes and have noted some significant differences in terms of deleterious cardiovascular effects, as can be observed in Table 1. Most of the deleterious effects of tobacco use are associated with the exposure to toxic compounds resulting from the combustion of tobacco [80]. One of the main reasons for the increased popularity of electronic cigarettes is due to the elimination of the said toxic compounds by replacing the combustion mechanism with a heating one. In addition, e-cigs are promoted as a great alternative to traditional smoking and portrayed as helping with smoking cessation [2]. 

Nicotine absorption determines the release of dopamine mainly in the mesolimbic area, the frontal cortex, the ventral segmental area of the midbrains, and the nucleus accumbens, which signals a pleasant experience, reduces anxiety and stress, and reinforces further consumption of nicotine. Due to this aspect, smoking cessation has similar withdrawal effects to other types of substance abuse, such as increased irritability, depressed state, and anxiety [43]. The degree of the symptoms is dependent on a sum of factors such as period of smoking, the concentration of nicotine product used, and the daily amount consumed. However, although nicotine in itself is not a major player in the development of smoking-related diseases compared to the other toxic substances that it is paired with, it is responsible for chronic tobacco consumption [81]. Nicotine consumption can increase blood pressure and heart rate, whereas ENDS use has an almost immediate effect on airway resistance, decreasing the power to draw air in [21].

Although electronic cigarettes have a significant decrease in terms of carcinogens, they are not completely harmless, as the e-liquid contains propylene glycol and vegetable glycerine, which, when heated, leads to the formation of toxic counterparts, such as formaldehyde, acrolein, and acetaldehyde. All three of these are highly dependent on temperature and are considered to be a risk for cardiovascular events [82]. On top of the liquid, the heating mechanism behind electronic cigarette poses the risk of exposure to toxic contaminants such as cadmium, chromium, lead, nickel, and silicates, with some of them having no biological purpose for humans, and being therefore toxic [3,4]. Even if they are generally in low concentrations, nanoparticles of heavy metals are of concern due to the ease of systemically absorption. Electronic cigarettes are capable of producing highly reactive free radicals [3,5].

The final stage of all the previously described pathological mechanisms is cardiotoxicity, a structural and functional heart damage caused by different classes of therapeutic agents, atmospheric or food factors, and is one of the most common causes of therapeutic failure, withdrawal, or disapproval of drugs, and increased incidence of cardiovascular disease unrelated to individual risk factors [39]. Cardiotoxicity can be irreversible, when necrosis or apoptosis of the myocardial cells occurs [41], or reversible, as in the case of short-term nicotine product consumption (or even long-term smoking in heavy smokers [3,42]).

Toxic heart disease is caused by a variety of mechanisms, depending on the etiologic agent involved. The clinical manifestations include paroxysmal or permanent arrhythmias, systolic and/or diastolic dysfunction, and heart failure [39,41].

Mitochondrial dysfunction is primum movens in the chain of events leading to cardiac impairment, and can be induced by inhibiting mitochondrial DNA replication, anomalies of the electron transport chain, or increased oxidative stress. Among the most common biomarkers of oxidative stress are malondialdehyde, prooxidant marker, and superoxide dismutase and glutathione peroxidase, antioxidant markers [41,42,83,84,85].

The detrimental effects of aerosols associated with ENDS and HTPs start at the endothelial level [86,87]. The oxidative stress reduces the bioavailability of nitric oxide which in turn plays a key role in endothelium dysfunction alongside peroxynitrite (produced by superoxide) which causes low-density lipoprotein oxidation and inflammatory cytokines [88]. A healthy endothelium is responsible for the production of vasodilators; however this function becomes impaired when the endothelium is injured (e.g., by cigarette smoke or toxic compounds found in ENDS and HTPs) leading to an imbalance between vasoconstrictors and vasodilators [89,90]. The damaged cells from the endothelium must be replaced in order to maintain vascular tone, an ability that is reduced in users of e-cigs and HTPs. Adding insult to injury, platelet activation, coagulation cascade stimulation, and impairment of anticoagulative fibrinolysis leads to vascular disease [91].

Pharmacological methods to counteract these effects in other clinical contexts include the administration of resveratrol, cytidine 5-diphosphocholine (CDP choline), or L-carnitine [58,92,93,94] and a recent study evaluated the role of products used in traditional Chinese medicine in combating cardiotoxicity caused by cytostatic therapy [95], part of which is based on the same mechanisms of tissue injury described above. The studies were performed on human subjects and proved the effectiveness of the aforementioned pharmacological products in improving oxidative stress, reducing platelet activation markers, and inflammatory response, with notable antihypertensive, lipid-lowering, and cardiac remodeling prevention effects of resveratrol [92].

**Table 1 healthcare-11-00491-t001:** Main differences in cardiac deleterious effects between the ENDS, HTPs and CCs.

Conventional Cigarettes	E-Cigarettes	“Heat Not Burn” Products	
	↓ blood pressure ↓ heart rate [96,97]	↓ systolic and diastolic blood pressure [98]	
	↑ mitral annulus diastolic velocity, diastolic strain rate ↓ isovolumetric relaxation time, myocardial performance [99]	similar effects on heart rate, blood pressure and risk of arterial stiffness [38]	
	No sign. increases in arterial stiffness, heart rate, blood pressure [100]		
↑ fluoreodeoxyglucose uptake in spleen and aortic tissue (indicating activation of the splenocardiac axis) [101]	↑ fluoreodeoxyglucose uptake in spleen and aortic tissue [101]	↓ vascular endothelial dysfunction ↓ oxidative stress than e-cigs and traditional cigs [65]	
	↓ blood pressure after switching to e-cigs for 52 weeks [102]		
no diff. in baseline heartrate and hemodynamics ↑ heart rate, blood pressure [103]	no diff. in baseline heartrate and hemodynamics [103]		
	↑ blood levels of endothelial progenitor cells [104]		
	↑ levels of soluble NOX2-derived peptide ↓ Nitric oxide bioavailability [104]		
↑ blood pressure after 5 min traditional smoking ⇔ ↑ blood pressure after 30 min. of e-cigs [105]			
	↑ blood pressure and heart rate in occasional tobacco smokers		
	↑ improvements in endothelial function, vascular stiffness after switching to e-cigs in acute smokers, but no effect for chronic smokers ↓ systolic blood pressure		

## 4. Conclusions

The pandemic of ENDS and HTPs use has documented cardiac adverse effects. Considering the data regarding the high percentage of young people and adolescents who use these devices, knowledge of the molecular and physiopathological mechanisms that are set in motion by the exposure to the toxins emanating from ENDS and HTPs can prove particularly useful in the management of those cases that reach the pediatric and/or the pediatric cardiology departments with subjective accusations of arrhythmias or with paraclinical evidence of cardiac and/or respiratory dysfunction. The purpose of the review will be achieved if the entire medical body that comes into contact with the pediatric age groups will follow and try to counteract the adverse effects of the use of ENDS and HTPs. It is also important for us to know the factors that predispose teenagers to the use of these devices, as well as the implementation of information programs to sensitize public opinion to this problem.

## Figures and Tables

**Figure 1 healthcare-11-00491-f001:**
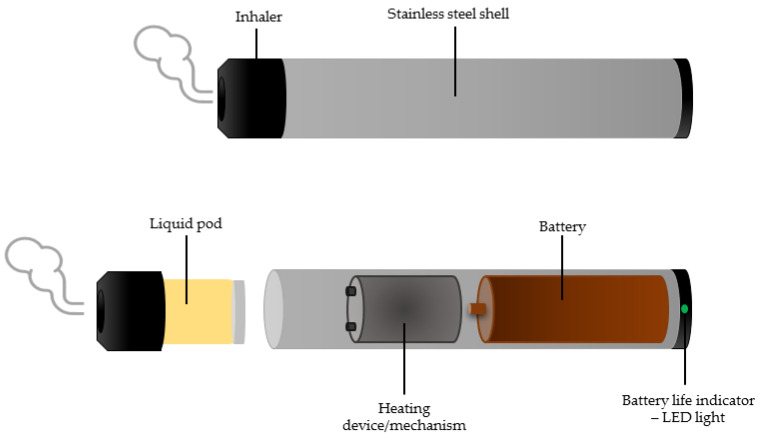
General structure of ENDS (e-cigarettes) comprised of a liquid pod, heating mechanism, battery and a battery life indicator using a LED light.

**Figure 2 healthcare-11-00491-f002:**
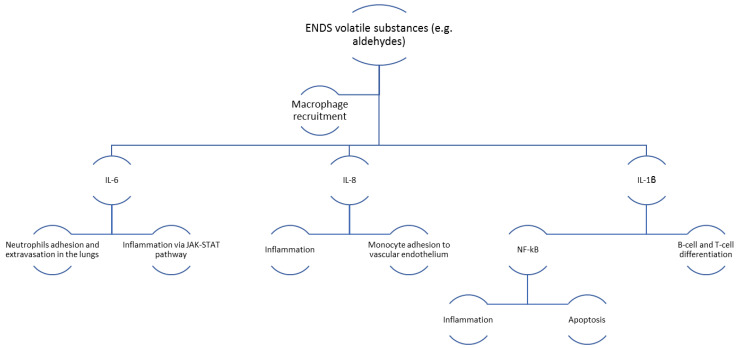
Schematic representation of the primary physiopathological mechanisms triggered by exposure to volatile substances from ENDS.

## Data Availability

Not applicable.

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
