# Peer review of "Cardiotoxicity of Electronic Cigarettes and Heat-Not-Burn Tobacco Products—A Problem for the Modern Pediatric Cardiologist"

_healthcare, 2023, doi:10.3390/healthcare11040491_

Round 1

Reviewer 1 Report

This article's theme is very interesting and acceptable.

However, the volume (description) of the article is very poor as review article.

For example, page 2-3, the description about the micro RNAs, it is important to describe in the details for readabilities. 

I think this manuscript is acceptable after major revision. 

Author Response

Dear Reviewer 1,

Thank you for your comments. Regarding your request of adding more information about micro RNAs, we have added extra information in the text (Lines 117-128). We hope this improves the quality of our paper.

Reviewer 2 Report

This short review describes the cardiovascular effects of electronic cigarettes and heat-not-burn tobacco products, focusing on the molecular basis, long-term cardiovascular effects, and acute complications. Overall, the review is well written, although the content is quite specialized and may be difficult for a pediatric cardiologist or other multi-disciplinary readers of this journal to understand. I have a few comments for the authors to improve the clarity and quality of their manuscript.

- As a reviewer within the fields of cardiovascular health and public health (non-physician), I found this review very focused on biochemical pathways and pathophysiology, which was unexpected based on the abstract. I suggest that the authors convey this information upfront in the abstract.

-Line 57 - What is the age range used for American adolescents? Are the prevalence estimates from the EU for an adolescent population - what was the age range used?

- The authors describe the prevalence of e-cigarette use internationally in the introduction. Quantitative evidence from epidemiological studies on the association between e-cigarette use and psychological, health care utilization, cardiovascular outcomes, and other outcomes would strengthen the introduction.

- Section 2 is quite complex and includes a lot of specialized biochemical terms. The authors may wish to include a figure that describes some of these molecular processes, which may simplify some of the messaging.

- The formatting in Table 1 makes the text difficult to interpret.

- Line 171 - "EDNS" should be "ENDS"

Author Response

Dear Reviewer 2,

Thank you for your insightful review. Regarding your comments, we have responded as follows:

  •  As a reviewer within the fields of cardiovascular health and public health (non-physician), I found this review very focused on biochemical pathways and pathophysiology, which was unexpected based on the abstract. I suggest that the authors convey this information upfront in the abstract.

We agree that our paper has focused on unmentioned aspects therefore we have added a clarifying extra information in the abstract (Lines 25-28).

  • Line 57 - What is the age range used for American adolescents? Are the prevalence estimates from the EU for an adolescent population - what was the age range used?

We apologize for missing out on this important detail. The age range used for American adolescents is highschool students whereas the age range for EU population was 11-17 years old (Lines 60-63).

  • The authors describe the prevalence of e-cigarette use internationally in the introduction. Quantitative evidence from epidemiological studies on the association between e-cigarette use and psychological, health care utilization, cardiovascular outcomes, and other outcomes would strengthen the introduction. Section 2 is quite complex and includes a lot of specialized biochemical terms. The authors may wish to include a figure that describes some of these molecular processes, which may simplify some of the messaging.

We have chosen to reply to these 2 comments in one as both focus on the outcomes of ENDS use. The extra information targetting the above mentioned comments can be found at Lines 64-67, Figure 1 and Lines 117-128. Hope these additions meet your criteria and going into the right directions.

  • The formatting in Table 1 makes the text difficult to interpret.

Thank you for your observation. We have changed the formatting of the tab;e and hopefully now it is easier to interpret. On each line the same factors are compared in correlation with each of the smoking type (conventional cigarettes, e-cigarettes, and heat-not-burn products).

  • Line 171 - "EDNS" should be "ENDS"

Corrected.

Round 2

Reviewer 1 Report

This paper was adequately modified, and I think this manuscript is acceptable in the present form.

Reviewer 2 Report

The authors have addressed my prior comments. I have no further suggestions.